# A Sustainable Approach for the Maintenance of Asphalt Pavement Construction

**Jelena Kilić Pamuković** * , **Katarina Rogulj, Daniela Dumanić and Nikša Jajac**

Faculty of Civil Engineering, Architecture and Geodesy, University of Split, Matice hrvatske 15, 21000 Split, Croatia; katarina.rogulj@gradst.hr (K.R.); daniela.dumanic@gradst.hr (D.D.); niksa.jajac@gradst.hr (N.J.)
* Correspondence: jkilic@gradst.hr

**Abstract:** The aim of this paper is to present a new decision support concept (DSC) related to the ever-growing problem of the maintenance of damaged asphalt pavements. In the process of defining a sustainable approach to resolving this problem, we found complexity in the different needs considering economic, social, and technical aspects. An additional contribution to the problem's complexity was the many road sections that need to be ranked based on their need for maintenance. The priority ranking was based on the multicriteria Preference Ranking Organization Method for Enrichment of Evaluations (PROMETHEE) method and the Analytic Hierarchy Processing (AHP) method. The DSC implementation contained the inclusion of relevant stakeholders and the definition of goals through identification of several different criteria and their weights. This approach to criteria determination provided a final ranking list of spatial units for maintenance, satisfying the needs of all stakeholders. The DSC presented in this paper was tested in the city of Split for the most important roads needing maintenance of asphalt pavements.

**Keywords:** decision support concept; maintenance; asphalt pavement; PROMETHEE; AHP

---

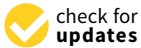



## 1. Introduction

Civil engineering deals not only with solving various individual technical problems but also with infrastructure systems such as transportation infrastructure, water supply infrastructure, etc. The quality of infrastructures impacts the quality of life of the people living in cities. Problems of traffic flow impact transportation infrastructures substantially. Recently, cities have been expanding rapidly, and the need to use motor vehicles is growing. As the number of vehicles increases in the existing traffic infrastructure, new problems emerge. Due to increased and continued traffic crossing, pavements are more easily damaged, and at critical locations, the question is how stable are they? On the other hand, the maintenance of pavements often does not follow this growth fast enough, which can cause pavement dilapidation. Studying this view for the reconstruction of necessary pavement sections and their maintenance, we found some existing literature. Jonson [1], in his book on asphalt pavement maintenance, discussed the importance of pavement preservation and preventive maintenance as well as presented techniques for dealing with a variety of problems and conditions. Specifications, and technical and special provisions were included for all treatment methods recommended in the handbook. The book by Jonson [1] is important because it provides the main paradigm of this paper. Namely, the focus of this paper is on the sustainable management of urban road infrastructure, particularly the sustainable management of asphalt pavements (as the most significant and important part) through the planning of preventative maintenance while the pavement is still in a relatively good condition (before it experiences structural problems—problems of insufficient subgrade capacity) and before its replacement is required. Sustainability as an important issue is integrated through the process of planning maintenance within the constrained available resources (financially, temporally, equipment, etc.) while maintaining a satisfactory quality of road infrastructure (technical aspects of maintenance on

infrastructure elements, such as road sections, as well their feasibility, etc.) and level of service (LOS) for end users (within the area under analysis). Rajagopal and George [2] explored the extent to which the timing and level of maintenance activities influence pavement conditions. Therein, mechanistic empirical models were developed to predict immediate changes in pavement conditions. Shahin and Kohn [3] presented the PAVER (The Pavement Maintenance Management System). The PAVER is designed to optimize the funds allocated for pavement maintenance and rehabilitation. The system includes procedures for dividing the pavement into manageable sections, for pavement condition rating and evaluation, for rational determination of maintenance needs and priorities, and so on. Additionally, they developed an economic analysis for the field implementing the PAVER [4]. A Markovian model to predict pavement deterioration with the inclusion of pavement improvement was used for an integrated pavement management system in the research of Abaza et al. [5]. Fwa et al. [6] developed a genetic-algorithm-based procedure to solve multi-objective network-level pavement maintenance programming problems. Cafiso et al. [7] presented a framework of alternatives in conflict ranked by multicriteria analysis. The analytic hierarchy process method (AHP) [8] was selected for pavement maintenance. The authors concluded that the maintenance of pavement ranking can distribute the budget more effectively than traditional economic priority settings. Another study used fuzzy AHP and the fuzzy technique to rank sections of pavement which need maintenance and rehabilitation [9]. Abu Dabous et al. [10], in their research, used multicriteria decision analysis techniques to rank and prioritize pavement sections for maintenance. They used a work breakdown structure to identify the decision elements. AHP and multi-attribute utility theory were used to rank and prioritize the network of pavement sections. They did not include the cost or budget allocation for maintenance of pavement sections. Zhu et al. [11] studied decision-making regarding asphalt pavement maintenance based on a life-cycle assessment and life-cycle cost analysis. The pavement maintenance, life cycle cost, and environmental impact were taken into account by building a decision-making index. Gao, Wu, and Feng [12] established a road pavement maintenance decision-making methodology taking into consideration the road state and traffic volume and integrating them into the evaluation criteria of the road technology condition. Decision-making models for the rehabilitation and reconstruction of asphalt pavement based on Rough Set Theory were proposed by [13,14]. Other studies used various methods to analyze pavement maintenance, including the VIKOR (VlseKriterijumska Optimizacija I KompromisnoResenje) method and the MACBETH (Measuring Attractiveness by a Categorical Based Evaluation Technique) method [15,16].

In the last few decades, the increasing traffic load has sped up the deterioration and ageing of pavements. Worn-out pavements lead to environmental, financial, and social issues, such as high expenses for vehicle repair, traffic delays, time delays, accidents, and fuel emissions. These factors create a great challenge for road managers to maintain the serviceability and functionality of the pavement. Hence, diverse maintenance and recovery activities are needed to counteract the negative influence of pavement deterioration. Classical maintenance and recovery technologies, such as in-place pavement recycling, wearing courses with very high reclaimed asphalt pavement contents, milling and filling, industrial waste and byproducts, are favorable, but the efficiency of these technologies relies on their implementation context [17]. Pavement maintenance is a strategy adopted to maintain, recover, or improve functionality of the pavement surface. It is applied to road pavements with acceptable structural strength and surface deterioration. When it comes to maintenance of road assets, determining the contradiction between the financial burden of maintenance and the level of service of highways is a major preeminence. Decisions in pavement maintenance are often typical multicriterial decisions that are influenced by various factors, such as policies, financial conditions, road detriment, and environment requirements [18]. In order to create a sustainable approach for maintenance of pavements, it is necessary to spot the most critical locations on roads which have to be repaired. Choosing those critical locations on pavements is a highly complex and socially sensitive process



considering various criteria for selection [19]. Nevertheless, it must be emphasized that, if the subgrade capacity is fully exceeded and the distresses identified in the pavement are related to structural deficiencies, then it is necessary to recycle the asphalt pavement completely. In such a case, preventive and maintenance treatments are not the right approaches, and corrective or emergency maintenance are necessary, which are not the focus of this paper. In the case of other types of asphalt pavement damage that are known as flexible pavement distresses (such as cracks, roughness, weathering, raveling, rutting, and bleeding), it is rational to provide repair. Maintenance that can be applied includes crack treatment, including crack repair with sealing (clean and seal, saw and seal, and rout and seal) or filling, and full and partial depth crack repair; surface treatments, which uses fog seals, seal coats, double chip seals, slurry seals, microsurfacing, and thin hot-mix overlays; and pothole patching and repair, using cold-mix asphalt, spray injection patching, hot-mix asphalt, and patching with slurry or microsurfacing material. Except the need to solve the above problem of choosing the most critical spots for repairing pavements, it is necessary to develop methods for planning and decision-making in troubleshooting. In fact, the timely and quality resolution of particular problems results in better quality systems as a whole. Therefore, an import aspect of civil engineering is the management of technical systems [20]. As an integral approach to management is of paramount importance, it is necessary to involve different stakeholders in decision-making to gather all the data and information necessary for solving this problem. This approach leads to better final choices between more and important solutions or decisions. With the rise in the necessity to solve this problem, consideration should be given to the public point of view and their need for road safety and for the best possible experience when using an essential road. Additionally, the technical aspects in repair and economics should be taken into account. It is important to consider whether an area of pavement is more or less damaged and whether the readiness for repair is at the required level. All of those aspects are characterized by a variety of technical and economic data and information and thus by the number of participants whose views need to be considered and appreciated. It is necessary to place these aspects in the context of the legal entities who are in charge of managing the transport infrastructure as well as their interaction with other participants or stakeholders. The complexity increases when all stakeholders are considered, and their opinions diverge because their needs are not equal in strength. This is based on different opinions of various stakeholders like citizen representatives, transportation experts, and economic experts. Citizen representatives request visual safeness on roads and demand a higher level of service on the most important streets. Transportation engineering experts have limitations in their choices of maintenance which covers the highest level of constructability that matches the highest technical standards. There are also conflicting goals of economic expert or government representatives based on budget restrictions and realization of needed documentation. Looking at only one aspect will result in the choice of a solution for that particular view. Therefore, it is necessary to find a solution to a sustainable-development concept which includes the opinions of all stakeholders or criteria for the decision [20]. A compromised solution should result in the most acceptable solutions for all.

Considering the complexity of the problem presented, different opinions of stakeholders for better resolving sustainable development and a naturally large number of critical locations on roads for maintenance, this paper will focus on the research problem of a sustainable approach for maintenance of asphalt pavements on main roads in the city of Split, Croatia. The main problem is to identify all critical spots and all relevant criteria from different aspects as well as their interrelationships for comparison. The aim of the paper is to model a new decision support concept (DSC) by decision-making tools such as multicriteria methods. Decision-making is the process of choosing among alternative solutions for the purpose of attaining a goal or a set of goals [21]. Generally, those supportive tools provide resolutions for well- and poorly structured planning decisions. Therefore, this can be useful and can enable sustainable and inclusive decision-making in planning of critical spots for the maintenance of pavements. The proposed model is based on the use

of multicriteria analysis methods, more accurately the Preference Ranking Organization Method for Enrichment Evaluation (PROMETHEE) [22] and Analytic Hierarchy Process (AHP). Those methods are useful for resolving such problems when considering a large number of critical spots, the level of conflict, and the diversity of the analyzed aspects. This approach relates to supporting planning processes and to the development of a specific plan which can be demonstrated and applied to other critical spots for the maintenance of pavement.

The authors use those methods in their research in civil engineering and in various fields of science. AHP and PROMETHEE are also used in researching ecology [23,24], geodesy [25–28], intelligent manufacturing [29], civil engineering [30,31], agronomy [32], software engineering [33], etc. Similar examples of using a multicriteria approach for urban transport management in the field of civil engineering as mentioned above can be found in the following works: introducing a multicriteria method for transportation investment planning [34], presenting a decision support system (DSS) approach to urban traffic management for the entire urban traffic system [35], and presenting fuzzy multicriteria ranking of urban transportation investment alternatives [36]. Additionally, the work in [37] and [38] aimed for multicriteria decision-making processes that include the AHP method for weighing a set of criteria. Studies [39–42] used the AHP method for asphalt pavement maintenance prioritization. Jajac et al. presented the decision support concept to urban infrastructure maintenance management [43] for managing the maintenance of city parking facilities [44]. An overview of the application methods of multicriteria analyses for making decisions about transport infrastructure can be found in Deluka-Tibljaš et al.'s paper [45].

In the city of Split, there are significant difficulties regarding damaged pavements and traffic flow increase, especially during the summer months due to tourism. Additionally, the government has a limited budget so resolving the issue of maintenance of asphalt pavements is complex. Since the focus of this research is on the improvement of pavement repair and maintenance, some traffic characteristics were analyzed. All stakeholders present their view on the most important, often conflicting, criteria and the criteria weights. According to the proposed DSC, the data were collected in 2018, and the model of multicriteria analysis is presented below.

## 2. Materials and Methods

Figure 1 shows the architecture of generic DSC for maintenance of asphalt pavements. Application of the concept begins with determination of the research problem including determination of the study area and selection the relevant stakeholders, who were divided into three groups:

- citizen representatives (9 representatives of city districts),
- experts in transport engineering (2 engineers from the utility company in charge of road maintenance; 2 engineers from the company that performs road maintenance works; 3 construction experts from the Faculty of Civil Engineering, Architecture, and Geodesy; 2 engineers from the Faculty of Civil Engineering, Architecture, and Geodesy; and 3 experts in methodology), and
- government representatives (representative of the utility company in charge of maintenance, deputy mayor in charge of infrastructure, and head of the department in charge of construction and infrastructure). Each group came up with common weighting criteria. The experts in charge of the methodology were tasked with explaining to all other stakeholders how the criteria were compared. Finally, the weights of the criteria were determined by the AHP method [8].

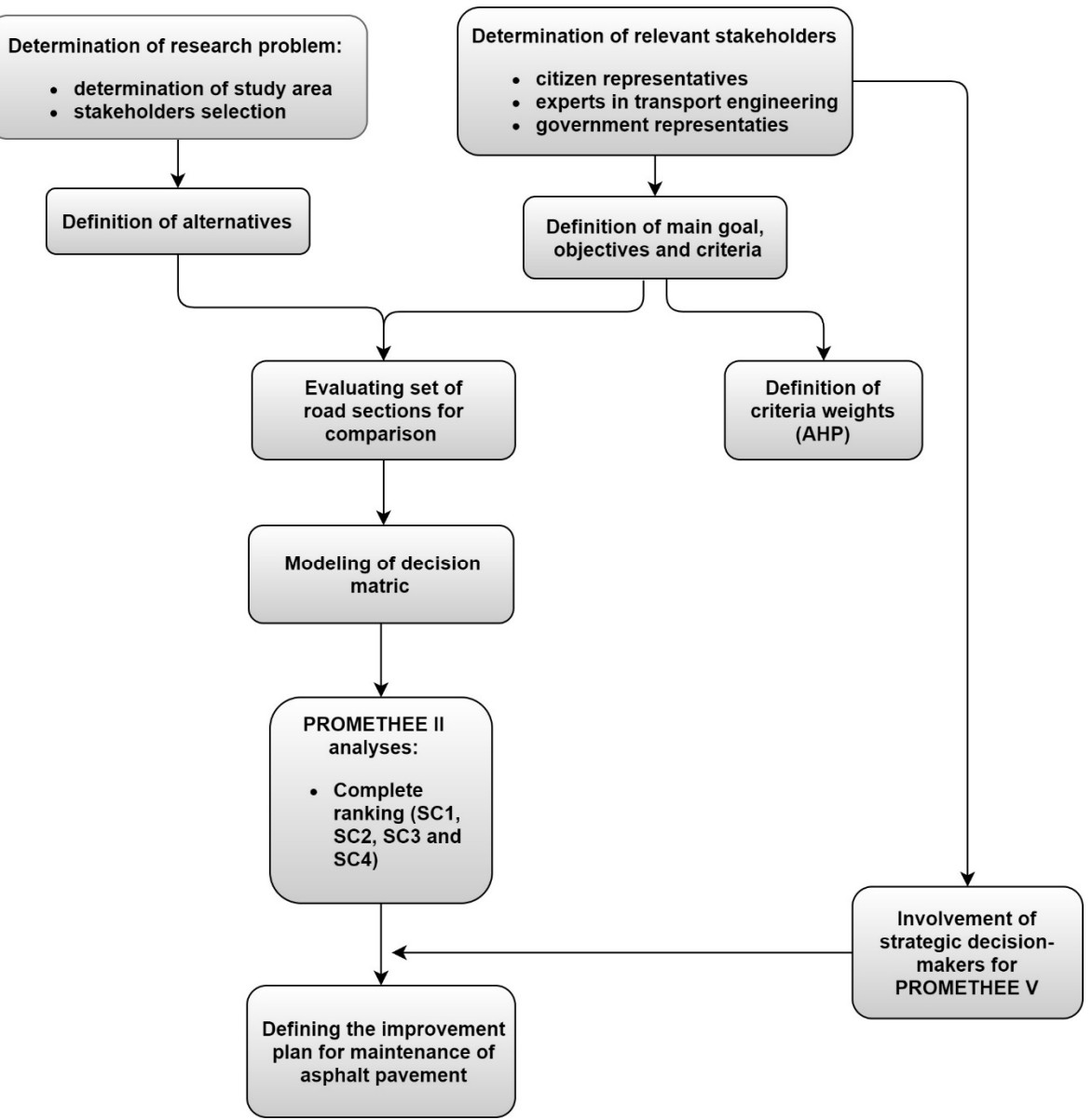

**Figure 1.** Concept for maintenance of asphalt pavements.

First, the alternatives were defined and the goal tree was formulated (goal hierarchy structure—GHS). GHS is composed of the main goal and the objectives and their criteria. All stakeholders are involved in that process. The main goal is defining a sustainable approach for maintenance of asphalt pavements, and it is at the top of the hierarchy. For its realization, it is necessary to determine supporting objectives which are subgoals of the main goal. The objectives were also divided in their supporting objectives, and it goes on like that until they become measurable and indivisible. At that point, the objectives were used as criteria.

The next step was identification of road sections in the study area which were relevant for comparison in order to reach the main goal. After the set of road sections was evaluated, it was possible to create a decision matrix which included all defined criteria. Selected road sections must be harmoniously decided upon by all stakeholders, and their opinions for criteria characteristics are expressed by criteria weights and selected preference functions. Criteria weights were determined for every group of stakeholders using the AHP method. A combination of the specified weights and criteria was presented as a strategy called scenario. Three scenarios were developed, one for each group of stakeholders. The fourth

scenario was based on an average value of all weights as a compromised view of the problem. All four scenarios were used for analysis and comparison of data by applying the methods PROMETHEE I and II. In the data analysis process, ranked lists of road sections (road sections are called actions) were studied and the results of the method analysis for the actions were compared for each scenario.

In the last step, strategic decision-makers were included in defining the improvement plan for maintenance of asphalt pavements when considering the obtained results of the analysis according to the created decision support concept and PROMETHEE V method for a compromised scenario. The results obtained by the PROMETHEE I and II methods provided insight into the relative relationship between individual alternative solutions. The final ranking list together with the plans, strategies, and the financial plan of the city of Split served as a basis for the definition of the annual maintenance plan of the asphalt pavement structure. Based on the above, certain constraints were introduced, which should primarily cover the spatial-functional and financial segments. Constraints were introduced by the final decision makers (in a specific case, they are representatives of local government units), and they were implemented using the PROMETHEE V method. The result was a set of road sections that need to be repaired on an annual basis. It should be emphasized that the proposed concept is applicable to other areas as well as to other elements of road infrastructure. First of all, the concept is flexible and it is easy and fast to make certain changes depending on the specific task. The changes most often refer to the last level of the goal hierarchical structure (GHS), which refers to the criteria used when comparing variant solutions. It is possible to add new or to amend and completely remove existing criteria in order to adapt the concept to the specific task, but it is especially important that decisions are made in the same way as in this process, involving all relevant stakeholders.

### 2.1. Multicriteria Methods

In the early 1980s, the methods and techniques of Multicriteria Decision-Making (MCDM) began to develop. Their basic application is in complex tasks, such as the problem of this research, which is characterized by a large number of spatial and other data and information, a large number of stakeholders and criteria to be considered. Based on conflicting criteria, the MCDM most often deals with ranking of alternative solutions identified in a specific task. To date, many MCDM methods have been developed and are being developed on a daily basis, with the most accepted being AHP, PROMETHEE, ELECTRE (ELimination Et Choix Traduisant la REalité), and TOPSIS (Technique for Order Preference by Similarity to the Ideal Solution). As part of this research, the AHP method for determining the importance of criteria and the PROMETHEE method for ranking alternative solutions will be used. Their brief descriptions are given below.

### 2.1.1. The AHP Method

The AHP method, as a multicriteria decision-making method, determines the ranking list according to the identified alternative solutions evaluated according to the defined criteria. Additionally, the method determines the importance factor (weight) according to pairwise comparisons of stakeholders involved in the decision-making process. Higher weights define a criterion of greater importance, while lower weights define a less important criterion. The final ranking is obtained by combining the weights of the criteria and the grade of alternative solutions. Precisely because of the weights that are a matter of stakeholders' subjective assessment, groups of stakeholders are usually formed according to the common preferences they share. The final criterion weight is always the averaged value of the weights defined by each group individually. Because of its simplicity in calculation as well as easy familiarization of the way the method works to stakeholders, the AHP is often chosen when it is necessary to mathematically define the different preferences of stakeholder groups.

To calculate the weights of different criteria, it is necessary to create a pairwise comparison matrix A. Matrix A has dimensions $n \times n$, where $n$ is the number of considered

evaluation criteria. Each number $a_{jk}$ in the matrix represents the importance of the $j$th criterion in relation to the $k$th criterion. Value $a_{jk}$ defines relationship between the $j$th and $k$th criteria:

-     $a_{jk} > 1$—$j$th criterion is more important than the $k$th criterion,
-     $a_{jk} < 1$—$j$th criterion is less important than the $k$th criterion,
-     $a_{jk} = 1$—$j$th and $k$th criteria are equally important.

The Saaty [8] numerical evaluation scale is used to measure the relative importance between criteria. After defining matrix A, it is necessary to determine its normalized shape ($A_{norm}$). When the sum of the entries in each column reaches the value of 1, each entry ($\overline{a_{jk}}$) of the matrix ($A_{norm}$) is computed as follows:

$$\overline{a_{jk}} = \frac{a_{jk}}{\sum_{l=1}^{m} a_{lk}} \tag{1}$$

The weight vector of criterion $w$ is equal to the arithmetic mean of each row in the normalized matrix:

$$w_j = \frac{\sum_{l=1}^{m} \overline{a_{jk}}}{m} \tag{2}$$

The matrix of the relative rankings is generated for each level of hierarchy bay pairwise comparison. The dimension of the matrix is determined based on the number of elements in each level. The vector of relative weight and maximum eigenvalue ($\lambda_{max}$) for each matrix are calculated after all matrices have been created.

To calculate the consistency ratio, it is necessary to calculate the consistency index (CI) of a $n \times n$ matrix:

$$CI = (\lambda_{max} - n)/(n - 1) \tag{3}$$

where $\lambda_{max}$ is the maximum eigenvalue of the matrix and $n$ is the matrix dimension.

The consistency ratio (CR) to validate comparisons is calculated as follows:

$$CR = CI/RI \tag{4}$$

where RI value is the random consistency index.

Depending on the dimensions of the matrix, an acceptable value of CR is determined (0.1 for matrices $n \geq 5$). Evaluation within the matrix is allowable if the CR value is equal to or less than the specified value.

### 2.1.2. The PROMETHEE Method

The PROMETHEE method is one of the best-known methods of multicriteria analysis developed with the intention of assisting decision makers in solving multicriteria decision problems. It was developed by J.P. Brans and B. Mareschal, and today, it is well accepted among decision makers because it is comprehensive and has the ability to determine results using simple ranking [46]. A comparison was performed, and ranking of different alternative solutions was simultaneously evaluated on the basis of several quantitative or qualitative criteria (attributes). It belongs to the class of so-called "outranking" methods that can be said to represent a compromise between an overly "poor" relation of dominance and the assumption that the decision-making utility function is known [47].

There are six types of preference functions proposed by the authors of the method [46]: usual criterion, U-shape criterion, V-shape criterion, level criterion, linear criterion, and Gaussian criterion. Each criterion is assigned one of the proposed types of preference functions and their relative importance (weight, weight) is determined. The preference index $\Pi$ as the weighted mean of the preference function $P_j$ can be defined by the following expression [42]:

$$\prod(A_i, A_j) = \frac{\sum_{j=1}^{n} P_j(A_i, A_j)}{\sum_{j=1}^{n} w_j} \tag{5}$$

where $P_j(A_i, A_j)$ represents the preference $A_i$ over $A_j$, and $w_j$ represents the weight of the $j$th criterion.

The preference index $\prod(A_i, A_j)$ represents the preference intensity of the decision maker of the variant solution $A_i$ over the solution $A_j$ simultaneously considering all the criteria. Since it is most often $\sum_{j=1}^{n} w_j$, expression (3) can be converted to the following form [46]:

$$\prod(A_i, A_j) = \prod(A_i, A_j) = \sum_{j=1}^{n} P_j(A_i, A_j) \tag{6}$$

Equally, the preference index $\prod(A_j, A_i)$ expresses how and with what intensity $A_j$ dominates over $A_i$ in relation to all criteria. Given the above, for each variant solution, two outranking flows can be defined [46]:

- output or positive ranking flow, which represents the sum of the values of all arcs coming out of the node (alternatives, activities) $A_i$ and expresses the measure of how much variant $A_i$ dominates over all other variants ($A_j \in A$) according to all criteria

$$\Phi^+(A_i) = \sum_{A_j \in A} \prod(A_i, A_j) \tag{7}$$

- input or negative ranking flow, which represents the sum of the values of all arcs entering node $A_i$ and expresses the measure of how much other variants dominate variant $A_i$ according to all criteria.

$$\Phi^-(A_i) = \sum_{A_j \in A} \prod(A_j, A_i) \tag{8}$$

By comparing the input and output flows, two complete rankings of the set of variants are obtained and their cross-section in the partial ranking gives the final ranking obtained by the PROMETHEE I method.

For the overall complete ranking of PROMETHEE II for the set of variants A, the net flow $\Phi$ can be calculated as the difference between positive and negative flows [46]:

$$\Phi(A_i) = \Phi^+(A_i) - \Phi^-(A_i) \tag{9}$$

The result of this step is a partial and complete ranking of road sections.

**PROMETHEE V**

If in certain cases it is necessary to determine a subset of variant solutions with respect to a set of constraints, the PROMETHEE V method is used, which is an extension of the PROMETHEE I and II methods. Bool variables are often used to solve constraint-related problems. If $\{A_i, i = 1, 2, \ldots, n\}$ is a set of possible variant solutions, then we have the following [48]:

$$x_i = \left\{ \begin{array}{l} 1, if\ A_i\ is\ selected \\ 0, if\ A_i\ is\ not\ selected \end{array} \right\} \tag{10}$$

The PROMETHEE V method can be observed through two phases or steps [48]:

Step 1—The multicriteria problem is first considered without the segmentation constraint solved by the PROMETHEE I and II methods. Complete ranking from the PROMETHEE II method was obtained with the calculated net flows $\{\Phi(A_i), i = 1, 2, \ldots, n\}$,

Step 2—Additional segmentation constraints are introduced taking into account linear programming $\{0, 1\}$:

$$\max\left\{ \sum_{i=1}^{k} \Phi(A_i) x_i \right\} \tag{11}$$

$$\sum_{i=1}^{n} \alpha_{p,i} x_i \cong \beta_p p = 1, 2, \ldots, P \tag{12}$$

$$\sum_{i \in S_r} \gamma_{q_r,i} x_i \cong \delta_{q_r}, q_r = 1, 2, \ldots, Q_r \qquad (13)$$

$$x_i \in \{0, 1\}, i = 1, 2, \ldots, n$$

where $\cong$ means =, $\leq$, or $\geq$, and $\alpha_{p,i}$ and $\gamma_{q_r,i}$ are the constraint coefficients. The coefficients of the objective function (9) are the net outranking flows, i.e., the higher the net flow, the better the variant solution. The linear programming solution $\{0, 1\}$ provides a subset of variant solutions that satisfy the constraints and ensures the highest possible net flow.

## 3. Results

Urbanization and the increasing number of vehicles that follow from the growth of a city population and its surroundings also affect the condition of pavements due to traffic flow. Under the influence of traffic flow and ecological impacts, pavements gradually lose capacity and degrade. Maintenance of pavements is a job that should be done regularly to keep it in the best possible state to ensure road safety. Hence, the main goal is to choose the best sustainable approach for maintenance of pavements because, in the city of Split and its surroundings, there are a lot of asphalt road sections with this problem. The study area was surveyed in detail, resulting in determination of the most important road sections which are used for validation of the proposed DSC (Figure 2).

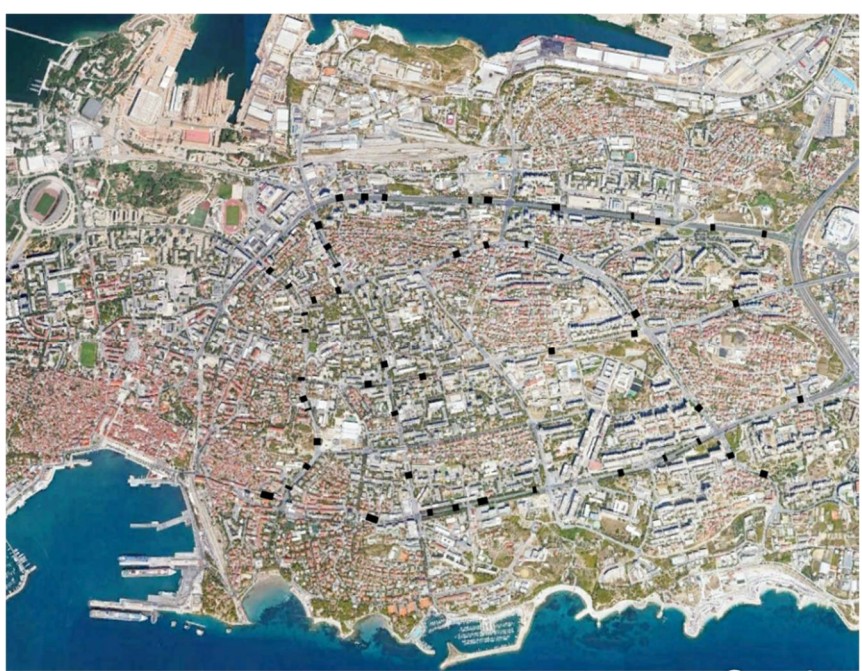

**Figure 2.** Road sections used for validation of decision support concept (DSC).

Fifty roads were chosen to assess if there was any type of damage along the length of section proportional. They included all types of roads in Split. The selected roads were county road Ulica Domovinskog rata and state road Poljička cesta, which connect Split to the surrounding area, and local roads Ulica Slobode, Dubrovačka ulica, Velebitska ulica, and Vukovarska ulica. Except for defining the study area, it is necessary to determinate adequate groups of stakeholders. Three groups of stakeholders were identified and organized: citizen representatives, experts in transport engineering, and government representatives. All of them were instructed and informed about the problem, and every group of stakeholders participated in defining the GHS (Figure 3).

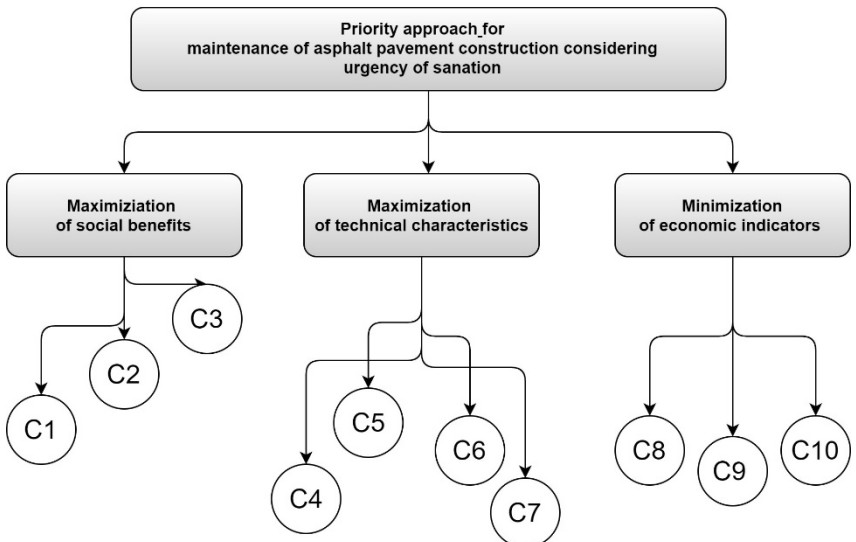

**Figure 3.** Goal hierarchy structure.

GHS begins with definition of the main goal: "A sustainable approach for maintenance of asphalt pavements". According to the "wish-list", which is created by all stakeholders, the first level of objectives contains maximization of social benefits, maximization of technical characteristics, and minimization of economic indicators. The next level of supporting objectives is the last level that represents criteria which are used for multicriteria analysis. Maximization of social benefits is divided into three criteria (C1–C3), maximization of technical characteristics is divided into four criteria (C4–C7), while minimization of economic indicators is divided into three criteria (C8–C10). There are a total of 10 criteria for evaluating alternative solutions (Figure 3), and their positions depend on the average value of criteria weights in a way that a criterion with a higher criteria weight is above a criterion with a lower value of criteria weight (getting those values will be explained later). It was decided that the form of the GHC would be a tree because it provides insight into the relative relationship of goals, subgoals, and criteria. The relative ratio was defined by the weights determined by each group of stakeholders separately using the AHP method.

Criteria names, their descriptions, the techniques used to evaluate the condition of road sections, and the selected preference functions are shown in Table 1. The choice of the preference function was based on the scientific and professional experiences of experts in solving similar problems. Prior to their final definition, it was necessary to consult representatives of stakeholder groups involved in the decision-making process. The values of the preference functions ranged from 0 to 1; values close to zero defined the high indifference of decision makers while values close to one defined their high preference. Strict preference was defined with a value of 1. Maxima and minima were searched for the five criteria equally. V was the shape function of preferences used for eight criteria, and U was the shape function used for the remaining two criteria.

**Table 1.** Criteria descriptions and preference functions.

| Criteria Label | Criteria Name | Short Description of Criteria and Technique for Evaluation Solutions | Preference | |
|---|---|---|---|---|
| | | | min/max | Function |
| C1 | Road categorization | Rating 1 for 1st road category to 5 for 5th road category | max | Usual |
| C2 | LOS | Level of service; rating 1 for LOS A to 6 for LOS F | max | Usual |
| C3 | Maximum level of safeness | Social rating for requisite for increasing sight distance, reducing traffic jams, increasing road equipment; grading 1 (worst)–10 (best) | max | Usual |
| C4 | Geometric characteristics | If there are already added layers for better capacity—1; if there are not—0 | max | V-shape |
| C5 | Visual rating | Experts rate visual condition of road by PSI (Present Serviceability Index by AASHO (American Association of State Highway and Transportation Officials) Road Test); ranges from 0 (impossible road) to 5 (perfect road) | min | V-shape |
| C6 | Subgrade capacity | Expresses in CBR (%) | min | Usual |
| C7 | Deflexion of pavement | If there is elastic character—0; if there is noticeable deflexion—1 | max | Usual |
| C8 | Cost rating of performing maintenance work | Cost of 2000 € for low cost (10 m needed for performing), cost of 3000 € for middle cost (20 m needed for performing), and cost of 5000 € for high cost (50 m needed for performing) | max | Usual |
| C9 | Level of readiness for maintenance activities | If there is no documentation—0; there is some documentation—1; there is all documentation—2; there is documentation plus building permits obtained—3 | min | V-shape |
| C10 | Minimum realization time | Expressed in days | min | Usual |

AASHO: American Association of State Highway and Transportation Officials.

Each group of stakeholders has its own strategy for multicriteria decision-making and, based on this each group, has its own scenario with their defined criteria weights according to the AHP method: Scenario 1 (SC1) belongs to citizen representatives, Scenario 2 (SC2) belongs to traffic experts, and Scenario 3 (SC3) belongs to government representatives. Scenario 4 (SC4) is defined as a set of compromised criteria weights based on the first three scenarios. The arithmetic mean was used because of the stakeholders' preference. For each scenario, the eigenvector ($\Omega$) and then maximum eigenvector ($\lambda_{max}$) were calculated using the basic principles of the AHP method. After that, the consistency ratio (CR) was defined for each SC as $CR \leq 0.10$. Each expert from each scenario compared the criteria, and then, common criteria weights of each scenario were defined. Table 2 gives an example of criteria weight determination of one expert from Scenario 1. Additionally, $\lambda_{max}$ and CR are determined as 10.24 and 0.03, respectively. All scenarios with their criteria weights are presented below in Table 3.

**Table 2.** Criteria weights and consistency ratio (CR) from citizen representatives' stakeholder (SC1).

| | C1 | C2 | C3 | C4 | C5 | C6 | C7 | C8 | C9 | C10 |
|---|---|---|---|---|---|---|---|---|---|---|
| **C1** | 1 | 1/2 | 2 | 1/2 | 1/2 | 3 | 1/2 | 2 | 1/2 | 2 |
| **C2** | 2 | 1 | 1/2 | 1/2 | 2 | 1/2 | 3 | 1/2 | 1/2 | 1/2 |
| **C3** | $\frac{1}{2}$ | 2 | 1 | 2 | 1/2 | 2 | 2 | 3 | 2 | 2 |
| **C4** | 2 | 2 | $\frac{1}{2}$ | 1 | 2 | 3 | 1/2 | 1/2 | 3 | $\frac{1}{2}$ |
| **C5** | 2 | $\frac{1}{2}$ | 2 | 1/2 | 1 | 1/3 | 1/2 | 1/2 | 1/2 | 1/2 |
| **C6** | 1/3 | 2 | $\frac{1}{2}$ | 1/3 | 3 | 1 | 2 | 3 | 2 | 2 |
| **C7** | 2 | 1/3 | $\frac{1}{2}$ | 2 | 2 | 1/2 | 1 | 1/2 | 1/2 | $\frac{1}{2}$ |
| **C8** | $\frac{1}{2}$ | 2 | 1/3 | 2 | 2 | 1/3 | 2 | 1 | 2 | $\frac{1}{2}$ |
| **C9** | 2 | 2 | $\frac{1}{2}$ | 1/3 | 2 | 1/2 | 1/2 | 1/2 | 1 | 2 |
| **C10** | 1/2 | 2 | $\frac{1}{2}$ | 2 | 2 | 1/2 | 2 | 2 | 1/2 | 1 |
| **Σ** | 12.83 | 14.33 | 8.33 | 11.17 | 17.00 | 11.67 | 15.50 | 13.50 | 12.50 | 11.50 |
| **w** | 0.11 | 0.08 | 0.14 | 0.12 | 0.07 | 0.12 | 0.07 | 0.10 | 0.10 | 0.10 |

**Table 3.** Criteria weights and CR for four SCs.

| SC | CR | $\lambda_{max}$ | | C1 | C2 | C3 | C4 | C5 | C6 | C7 | C8 | C9 | C10 |
|----|----|----|----|----|----|----|----|----|----|----|----|----|----|
| SC1 | 0.070 | 10.940 | Ω | 4.13 | 14.30 | 20.10 | 0.60 | 4.13 | 2.40 | 0.60 | 0.60 | 0.60 | 2.40 |
| | | | w | 0.125 | 0.25 | 0.30 | 0.025 | 0.125 | 0.05 | 0.025 | 0.025 | 0.025 | 0.05 |
| SC2 | 0.063 | 10.845 | Ω | 6.50 | 9.30 | 18.80 | 9.30 | 9.30 | 18.80 | 6.50 | 1.90 | 3.70 | 3.70 |
| | | | w | 0.10 | 0.125 | 0.15 | 0.125 | 0.125 | 0.15 | 0.10 | 0.025 | 0.05 | 0.05 |
| SC3 | 0.044 | 10.585 | Ω | 3.70 | 6.50 | 18.80 | 1.90 | 6.50 | 3.70 | 1.90 | 1.80 | 18.80 | 18.80 |
| | | | w | 0.05 | 0.10 | 0.15 | 0.025 | 0.10 | 0.05 | 0.025 | 0.20 | 0.15 | 0.15 |
| SC4 | 0.079 | 11.061 | Ω | 9.30 | 15.60 | 21.10 | 3.90 | 10.90 | 6.20 | 3.70 | 6.20 | 4.40 | 6.20 |
| | | | w | 0.10 | 0.15 | 0.20 | 0.058 | 0.118 | 0.083 | 0.05 | 0.083 | 0.075 | 0.083 |

After calculating the compromise weights, C3 and C5 were determined as the most important criteria according to their weights, those with the highest values. Criterion C3 refers to the maximum level of safeness, while within C5, the visual rating was determined. Given their description, their importance in relative relation to the other criteria is fully justified. The least important criterion is C4, i.e., the criterion by which the geometric characteristics of alternatives were evaluated.

The decision matrix consist of 10 columns which represent criteria and 50 rows which represent alternative solutions (road sections). Each row gives the evaluation of one road section through 10 criteria, while each column gives an evaluation of all road sections with regard to one criterion. The analysis was performed using a software solution Visual PROMETHEE [49]. The software enables entry of the evaluated road sections according to the defined criteria, the weight of all criteria, as well as the preferences of stakeholders expressed through the minima and maxima, and the preference function. Part of the decision matrix is shown in Table 4.

**Table 4.** Part of decision matrix.

| Criteria Actions | C1 | C2 | C3 | C4 | C5 | C6 | C7 | C8 | C9 | C10 |
|----|----|----|----|----|----|----|----|----|----|----|
| action1 | 5 | 6 | 5 | yes | 3 | 14.4 | yes | 2000 | 2 | 8 |
| action2 | 5 | 6 | 5 | yes | 3 | 11.1 | yes | 2000 | 2 | 8 |
| action3 | 5 | 5 | 6 | yes | 3 | 5.3 | yes | 5000 | 2 | 20 |
| action21 | 4 | 6 | 8 | yes | 3 | 16.8 | yes | 2000 | 3 | 1 |
| action22 | 4 | 5 | 8 | no | 3 | 34.4 | yes | 2000 | 3 | 1 |
| action31 | 3 | 5 | 9 | yes | 4 | 29.8 | no | 1000 | 3 | 1 |
| action32 | 3 | 4 | 8 | yes | 3 | 8.7 | yes | 2000 | 3 | 2 |
| action41 | 2 | 5 | 8 | no | 3 | 12.3 | no | 2000 | 0 | 3 |
| action42 | 2 | 4 | 7 | no | 2 | 8.6 | yes | 1000 | 0 | 1 |
| action43 | 2 | 3 | 8 | yes | 3 | 18.8 | no | 1000 | 2 | 3 |

After determination of a decision matrix, the software Visual PROMETHEE [34] was applied for comparison and processing of all data for each scenario. A complete ranking of road sections was established by using multicriteria method PROMETHEE II. This method provides ranking by mutual comparison of all road sections for every criterion and the weight given by stakeholder opinions. Table 5 shows the obtained ranking list of road sections and their positive, negative, and complete (the difference between positive and negative) Phi net flows ($\Phi$). Phi net flow represents preferences among road sections in the way that a higher value for a certain road section means that alternatives express a greater need for repair then the other in the set. Positive and negative values of the complete Phi net flows show how certain a road section is better or worse than the other (26 road sections have a positive values of Phi net flow, while 24 have a negative value, which indicates their weaker status in regards to the first 26 alternative solutions).

**Table 5.** Net flow complete ranking (Preference Ranking Organization Method for Enrichment of Evaluations (PROMETHEE) II method).

| Rank | Tag | Phi | Phi+ | Phi− | Rank | Tag | Phi | Phi+ | Phi− |
|---|---|---|---|---|---|---|---|---|---|
| 1 | S19 | 0.392 | 0.492 | 0.1 | 26 | S10 | 0.0123 | 0.2513 | 0.239 |
| 2 | S30 | 0.3594 | 0.4734 | 0.1139 | 27 | S1 | −0.007 | 0.3209 | 0.3279 |
| 3 | S20 | 0.3321 | 0.4447 | 0.1126 | 28 | S46 | −0.0306 | 0.3085 | 0.3391 |
| 4 | S12 | 0.311 | 0.4558 | 0.1448 | 29 | S41 | −0.0427 | 0.2834 | 0.3261 |
| 5 | S36 | 0.3107 | 0.4589 | 0.1482 | 30 | S9 | −0.0596 | 0.271 | 0.3306 |
| 6 | S18 | 0.3029 | 0.4478 | 0.1449 | 31 | S17 | −0.0739 | 0.2546 | 0.3285 |
| 7 | S35 | 0.2477 | 0.4014 | 0.1537 | 32 | S14 | −0.0797 | 0.2036 | 0.2833 |
| 8 | S13 | 0.2112 | 0.3978 | 0.1866 | 33 | S33 | −0.1031 | 0.2181 | 0.3212 |
| 9 | S6 | 0.2015 | 0.4039 | 0.2024 | 34 | S32 | −0.1127 | 0.208 | 0.3207 |
| 10 | S48 | 0.1967 | 0.4177 | 0.221 | 35 | S15 | −0.1163 | 0.1872 | 0.3035 |
| 11 | S34 | 0.1933 | 0.3612 | 0.1679 | 36 | S40 | −0.1323 | 0.2331 | 0.3654 |
| 12 | S28 | 0.1773 | 0.4135 | 0.2362 | 37 | S43 | −0.137 | 0.2041 | 0.3411 |
| 13 | S3 | 0.1716 | 0.3974 | 0.2258 | 38 | S24 | −0.1444 | 0.1932 | 0.3377 |
| 14 | S29 | 0.1622 | 0.3886 | 0.2264 | 39 | S22 | −0.1726 | 0.1899 | 0.3625 |
| 15 | S11 | 0.1454 | 0.3544 | 0.209 | 40 | S44 | −0.1754 | 0.1766 | 0.352 |
| 16 | S38 | 0.1316 | 0.3638 | 0.2322 | 41 | S39 | −0.2031 | 0.1785 | 0.3816 |
| 17 | S45 | 0.1307 | 0.372 | 0.2413 | 42 | S25 | −0.2124 | 0.151 | 0.3634 |
| 18 | S42 | 0.0982 | 0.3559 | 0.2577 | 43 | S16 | −0.2202 | 0.1529 | 0.3731 |
| 19 | S46 | 0.0671 | 0.3412 | 0.274 | 44 | S4 | −0.2223 | 0.2176 | 0.4399 |
| 20 | S23 | 0.0348 | 0.3043 | 0.2695 | 45 | S8 | −0.2472 | 0.1298 | 0.377 |
| 21 | S7 | 0.0292 | 0.3274 | 0.2981 | 46 | S26 | −0.2858 | 0.1228 | 0.4086 |
| 22 | S37 | 0.0252 | 0.2792 | 0.2539 | 47 | S50 | −0.3025 | 0.2105 | 0.513 |
| 23 | S27 | 0.021 | 0.2929 | 0.2719 | 48 | S21 | −0.3216 | 0.1289 | 0.4505 |
| 24 | S49 | 0.0171 | 0.3437 | 0.3266 | 49 | S5 | −0.3682 | 0.1291 | 0.4973 |
| 25 | S2 | 0.0165 | 0.329 | 0.3125 | 50 | S31 | −0.528 | 0.0702 | 0.5982 |

Graphical presentation of the results is shown in Figure 4, which provides insight into the net flow results and PROMETHEE II complete priority ranking for the compromised scenario.

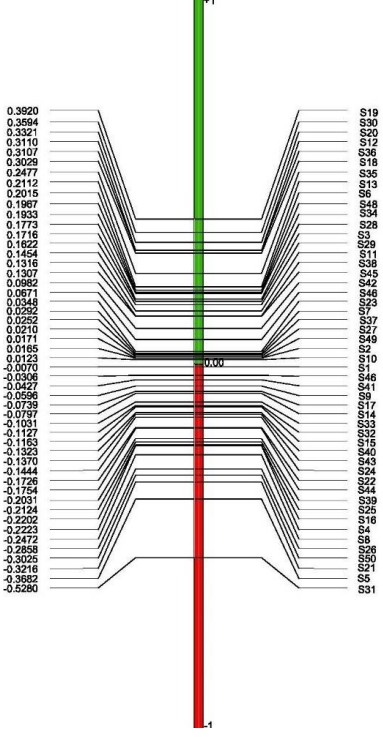

**Figure 4.** Graphical presentation of net flow results and PROMETHEE II complete ranking.

From the results, it can be seen that S19 has the highest net flow and that S31 has the lowest. This means that S19 is the element with the worst conditions and has the higher priority for repair. On the other hand, road section S31 represents the best conditions of a road section and a minimal need for repair. It can be observed that the values for the road sections are evenly distributed in the numerical direction related to the Phi net flow values (green represents positive values, while red refers to negative values). Spatial dislocation is expressed only by the last road section (the road section with the lowest value of Phi net flow—S31). Although the differences in Phi net flow values for road sections are small, the difference between the first and the last road sections is significant, with an absolute value of net flow $\Phi = 0.917$.

The next step is constraints definition, and it is conducted only by investors. These constraints are defined by 0–1 integer linear programming (as a set of linear equations and/or inequalities). In theoretical terms, there are three types of constraints, two of which are defined as part of this research. Some of them are universally applicable such as the financial constraint, while others refer to spatial analyzes for the purpose of uniform and equitable development of the analyzed area. The first is almost always related to finance as the most important limiting factor. The equations are defined in such a way that the left side consists of net flows of road sections (this part refers to alternatives/locations) while the right part is equal to the financial amount defined by the investment plan or decision of the decision-makers. The second type of constraints is related to the spatial-functional segment. These formulas describe a specific constraint arising from the desire to meet strategic goals. Spatial-functional formulas serve to define the constraints that seek to meet the strategic goal of uniformly maintaining the road infrastructure in the entire area of urban infrastructure under analysis. How many formulas will be defined for the spatial-functional goals depends on the decisions of the decision-makers on how much detail is needed for a uniform approach to road infrastructure maintenance. The third type of formula refers to explicit goals that need to be improved due to plans, regulations, and higher-order goals. When including a formula of this type, locations that are of explicit importance to decision makers are automatically selected. This approach is extremely important in practical terms because it allows for the overlap of road section ranking with the strategies and plans of local governments as well as the preferences of their representatives as ultimate decision-makers. The result is given in the form of annual sets of road sections that need to be repaired.

Within this research, one financial constraint was defined, while for the purpose of uniform development, the research area was divided into four subarea and, in relation to the above, four spatial-functional constraints were defined. As stated, one constraint within this set is related to the available financial resources (5,000,000 EUR) for next investment cycle. The four nonfinancial constraints (related to functional and spatial aspects of the analyzed problem) are as follows: 1st, at least two road sections from the 5th category; 2nd, at least one road section from the 4th category; 3rd, at least three road sections from the 3rd category; and 4th, at least two road sections from the 2nd category. The PROMETHEE V method is used for introduction of these 5 constraints into the investment planning process only by an investor. The goal function and above specified constraints which are used are shown below.

The final goal function is as follows:

$$Max \sum_{j=1}^{50} \Phi_j \, x_j \, ,$$

$$
\begin{aligned}
j &= 1,2,3,\ldots,50 \, ; \; Max\{0.3920x_1 + 0.3594x_2 + 0.3321x_3 + 0.3110x_4 + 0.3107x_5 + 0.3029x_6 \\
&+0.2477x_7 + 0.2112x_8 + 0.2015x_9 + 0.1967x_{10} + 0.1933x_{11} + 0.1773x_{12} + 0.1716x_{13} \\
&+0.1622x_{14} + 0.1454x_{15} + 0.1316x_{16} + 0.1307x_{17} + 0.0982x_{18} + 0.0671x_{19} + 0.0348x_{20} \\
&+0.0292x_{21} + 0.0252x_{22} + 0.0219x_{23} + 0.0171x_{24} + 0.0165x_{25} + 0.0123x_{26} - 0.0070x_{27} \\
&-0.0306x_{28} - 0.0427x_{29} - 0.0596x_{30} - 0.0739x_{31} - 0.0797x_{32} - 0.1031x_{33} - 0.1127x_{34} \\
&-0.1163x_{35} - 0.1323x_{36} - 0.1370x_{37} - 0.1444x_{38} - 0.1726x_{39} - 0.1754x_{40} - 0.2031x_{41} \\
&-0.2454x_{42} - 0.2495x_{43} - 0.2499x_{44} - 0.3011x_{45} - 0.3461x_{46} - 0.3497x_{47} - 0.3726x_{48} \\
&-0.3749x_{49} - 0.4595x_{50}\}
\end{aligned}
$$

(14)

The nonfinancial constraints are as follows:

$$x_1 + x_2 + x_3 + x_4 + x_5 + x_6 + x_7 + x_8 + x_9 + x_{10} + x_{11} + x_{12} + x_{13} + x_{14} + x_{15} + x_{16} + x_{17} + x_{18} + x_{19} + x_{20} \geq 2 \quad (15)$$

$$x_{21} + x_{22} + x_{23} + x_{24} + x_{25} + x_{26} + x_{27} + x_{28} + x_{29} + x_{30} \geq 1 \quad (16)$$

$$x_{31} + x_{32} + x_{33} + x_{34} + x_{35} + x_{36} + x_{37} + x_{38} + x_{39} + x_{40} \geq 3 \quad (17)$$

$$x_{41} + x_{42} + x_{43} + x_{44} + x_{45} + x_{46} + x_{47} + x_{48} + x_{49} + x_{50} \geq 2 \quad (18)$$

The financial constraint is as follows:

$$
\begin{aligned}
&0.3920x_1 + 0.3594x_2 + 0.3321x_3 + 0.3110x_4 + 0.3107x_5 + 0.3029x_6 + 0.2477x_7 + 0.2112x_8 + 0.2015x_9 \\
&+0.1967x_{10} + 0.1933x_{11} + 0.1773x_{12} + 0.1716x_{13} + 0.1622x_{14} + 0.1454x_{15} + 0.1316x_{16} \\
&+0.1307x_{17} + 0.0982x_{18} + 0.0671x_{19} + 0.0348x_{20} + 0.0292x_{21} + 0.0252x_{22} + 0.0219x_{23} \\
&+0.0171x_{24} + 0.0165x_{25} + 0.0123x_{26} - 0.0070x_{27} - 0.0306x_{28} - 0.0427x_{29} - 0.0596x_{30} \\
&-0.0739x_{31} - 0.0797x_{32} - 0.1031x_{33} - 0.1127x_{34} - 0.1163x_{35} - 0.1323x_{36} - 0.1370x_{37} \\
&-0.1444x_{38} - 0.1726x_{39} - 0.1754x_{40} - 0.2031x_{41} - 0.2454x_{42} - 0.2495x_{43} - 0.2499x_{44} \\
&-0.3011x_{45} - 0.3461x_{46} - 0.3497x_{47} - 0.3726x_{48} - 0.3749x_{49} - 0.4595x_{50} \leq 50000
\end{aligned}
\quad (19)
$$

Repeat usage of DSC for each of the following investment cycles (until all road sections are repaired) is recommended due to constant change in the project environment. Each time, a new set of road sections for repair is determined and it stands for one activity within the investment plan. According to the results of the PROMETHEE V method presented within Table 6, 11 of 50 analyzed road sections should be included in the investment plan for the first investment cycle.

**Table 6.** PROMETHEE V results—1st set of the investment plan.

| The First Set of Road Section | |
| --- | --- |
| S19 | S48 |
| S20 | S45 |
| S30 | S12 |
| S35 | S18 |
| S36 | S13 |
| S34 | |

## 4. Conclusions

With the aim of creating a decision support system for choosing the most important spatial units for maintenance of asphalt pavements, the proposed concept shows that the presented complexity can be appropriately reduced. Application of this concept makes it possible for the methods and data to be properly used. The advantages of this approach that is based on multicriteria analyses are shown through the inclusion of all relevant stakeholders. In this way, different opinions and attitudes towards finding a compromised solution to the specific problem are taken into account, which increases the quality of the planning process by avoiding mistrust and unfair preferences. The used methodology allows road-infrastructure ranking according to the requirements for improvement of their conditions. Applied to the road infrastructure of the city of Split, it seems to function well, and it can be used for any other road infrastructure.

Compared with existing decision-making models such as [11,12], it is obvious that these methods did not used the multicriterial approach to evaluate and analyze each alternative according to the defined criteria. No selection and division of relevant stakeholders was made, as a crucial factor in the whole decision-making process. Finally, the actions are mentioned, but their priority in the maintenance is not demonstrated and included in the overall outranking process. As for studies [13,14], only two activities are included in asphalt pavement maintenance under the Rough Set Theory in the decision-making procedure: the first evaluated the effect of failure on the intended function of the pavement, and the second is a visual inspection determining the condition of the pavement and the

problems that cause this condition. The methodologies are only based on the condition of the pavement, and no additional attention was given to other parameters that also have a notable impact on the whole construction, such as capacity, stability, safety, functionality, and costs.

Applying the PROMETHEE method for priority ranking and introducing constraints and the AHP method for determining criteria weights is a good basis for improvement in the planning process when taking into account social, technical, and economic aspects. The use of multicriteria methods within the DSC increased the reliability and objectivity in the adoption of annual plans for the maintenance of asphalt pavement constructions. The proposed concept was validated in the city of Split, where a set of 50 alternative solutions (road sections) was defined. The goal was to define priority ranking of the road sections that need to be repaired. Given that the main limiting factor is the financial nature and that it is necessary to meet the condition of uniform development of all parts of the city, restrictions were introduced: one financial limit defining the annual amount in the city budget for maintenance of asphalt pavement and four spatial-functional constraints with respect to the spatial division of the research area. The final result is an annual plan for maintenance of asphalt pavement construction which in this case consists of 11 road sections for research area. The proposed concept is unique and, above all, easy to use because it allows for simultaneous analysis of a large amount of different data and information with the introduction of stakeholder preferences with different attitudes and desires. In addition to all its advantages, certain disadvantages have been identified that will serve as a basis for future research. The reason for the length of this procedure is primarily in the method of data collection as well as the methods of assessing asphalt pavement. For full development of this concept in the direction of its expansion into a system that will also include other elements of road infrastructure, it is first necessary to automate the data collection process, using, e.g., a road infrastructure safety mapping system using georeferenced video. In order to assess the condition of asphalt pavements, expert systems and neural networks should be included in the future, which would speed up the process on the one hand while eliminating the subjective character of experts involved in solving a specific task on the other hand. The proposed concept is flexible and therefore more widely applicable, whether it is another case study or another element of road infrastructure. The above can be achieved by structural changes at the last level of GHS (criteria level) that includes adding new or amending and completely removing existing criteria in order to adapt the concept to the specific task.

**Author Contributions:** Conceptualization, J.K.P., K.R., D.D., and N.J.; methodology, J.K.P., K.R., D.D., and N.J.; software, J.K.P., K.R., D.D., and N.J.; validation, J.K.P., K.R., D.D., and N.J.; formal analysis, J.K.P., K.R., D.D., and N.J.; investigation, J.K.P., K.R., D.D., and N.J.; resources, J.K.P., K.R., D.D., and N.J.; data curation, J.K.P., K.R., D.D., and N.J.; writing—original draft preparation, J.K.P., K.R., D.D., and N.J.; writing—review and editing, J.K.P., K.R., D.D., and N.J.; visualization, J.K.P., K.R., D.D., and N.J.; supervision, J.K.P., K.R., D.D., and N.J.; project administration, J.K.P., K.R., D.D., and N.J.; funding acquisition, J.K.P., K.R., D.D., and N.J. All authors have read and agreed to the published version of the manuscript.

**Funding:** This research received no external funding.

**Institutional Review Board Statement:** Not applicable.

**Informed Consent Statement:** Not applicable.

**Data Availability Statement:** Data available on request due to restrictions eg privacy or ethical. The data presented in this study are available on request from the corresponding author. The data are not publicly available due to further research to be published.

**Acknowledgments:** This research is partially supported through project KK.01.1.1.02.0027, a project co-financed by the Croatian Government and the European Union through the European Regional Development Fund—the Competitiveness and Cohesion Operational Programme.

**Conflicts of Interest:** The authors declare no conflict of interest.

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
