# Peer review of "A Sustainable Approach for the Maintenance of Asphalt Pavement Construction"

_sustainability, doi:10.3390/su13010109_

Round 1

Reviewer 1 Report

The article submitted for review solves the relevant problem of quality assessment and improvement asphalt pavement with damages. To solve it, the authors use the known multiple criteria decision making (MCDM) methods PROMETHEE and AHP. A new decision support concept (DSC) has been introduced to assess economic, social or technical aspects.

The shortcomings noted in the work, which the authors of the article must take into account:

  1. The introduction does not contain an analysis of the principles of asphalt pavement maintenance and the technologies used. It would be interesting to know the Authors' opinion at what extent of asphalt pavement damage it is rational to repair, and when it is necessary to recycle it completely.
  2. How does asphalt pavement maintenance relate to sustainability in this work?
  3. The MCDM methods used in the work are based on the determination of the subjective normalized weights of the criteria and the prioritization of alternatives in the calculations. It seems unusual that the most important formulas of AHP and PROMETHEE methods are not presented in the work.
  4. It is necessary to specify whether the criteria weights were calculated from a single expert or from pairwise matrices completed by a group of experts. In all cases, it is necessary to calculate the eigenvectors “Omega” the largest eigenvalue λmax, and the consistency ratio (C.R.) of each expert completed matrix criteria to show whether the matrix is ​​consistent and whether the transitivity condition has been met. If the criteria weights have been determined from the average of the opinions of the expert group, it is also necessary to prove the consistency of their opinions by calculating, for example, the coefficient of concordance W using the Kendall rank correlation method. These calculations would show that the results of the study are reliable and that the presented methodology could be used by other researchers.
  5. The hierarchical structure presented in Figure 3 consists of a goal (the first level), three main criteria (the second level) and three to four sub-criteria (the third level). In Table 2, the weights of the ten sub-criteria for each of the four scenarios are presented as a percentage in a hierarchically unstructured form. It is not clear, by which way were they calculated? (PS. The integer part of the numbers must be separated by a dot, but not a comma).
  6. The calculation data (Table 2, 3, 4 and Fig. 4) are insufficiently analyzed in the text. Fig. 4. is not understandable to the reader because the legends are not visible (they are not legible). In formulas of the final goal function (1) and financial constraint (6) the commas must be replaced by dots. It is not enough sufficient explained how the result was obtained according the functions and the significance of this result both for science and practice was not revealed as well. Is the result worldwide universal, or is it only suitable for the city of Split?
  7. Conclusion is formulated due to insufficient disclosure of data obtained by calculations (AHP and PROMETHEE methods). They present the advantages, benefits and possibilities of applying the new DSC concept. However, the specialists of several asphalt pavement maintenance institutions will be able to use it in their activities to a limited extent.
  8. References are not presented in accordance with the requirements of the journal, i. e. the No of vol., issue, pp. and doi. are not marked at all in many publications of Reference list.

Author Response

Thank you very much for provision of remarks and suggestions. The authors hope that corrections and further explanations which are provided according to your comments and suggestions improved quality of the manuscript up to the satisfactory level it can be accepted for publication in Sustainability.

1.The introduction does not contain an analysis of the principles of asphalt pavement maintenance and the technologies used. It would be interesting to know the Authors' opinion at what extent of asphalt pavement damage it is rational to repair, and when it is necessary to recycle it completely.

Thank you very much for this comment because it pointed out to the authors the need for a more detailed presentation of this issue within the paper. Authors thought it was enough (in the Introduction chapter) to point out to the reference that cover the subject issue (Jonson [1]), because these (“the principles of asphalt pavement maintenance and the technologies used”) are generally known topics. Based on the above-mentioned and reviewer’s comment, several additional sentences are provided in the Introduction chapter to describe the above in more details.

In text: “Nevertheless it must be emphasised that if subgrade capacity is fully exceeded and the distresses identified in a pavement are related to structural deficiencies then it is necessary to recycle asphalt pavement completely. In such case, preventive maintenance treatment activities are not right approach. Then we are talking about corrective or emergency maintenance activities which are not in focus of this paper. In case of other asphalt pavement damages types that are known as flexible pavement distresses (such as cracking, roughness, weathering, ravelling, rutting and bleeding) it is rational to provide repairing. Maintenance activities/treatments that can be applied are: crack treatments - crack repair with sealing (clean and seal, saw and seal, rout and seal) or filling, full and partial depth crack repair; surface treatments – fog seal, seal coat, double chip seal, slurry seal, microsurfacing, thin hot-mix overlays; pothole patching and repair – cold-mix asphalt, spray injection patching, hot-mix asphalt, patching with slurry or microsurfacing material.”

2.How does asphalt pavement maintenance relate to sustainability in this work?

Thank you very much for this comment that helped us to see the need to explicitly show the relation between the management of asphalt pavements and sustainability within paper.

In text: "The paper of Jonson [1] is important because it provides the main paradigm of this paper. Namely, the focus of this paper is in sustainable management of urban road infrastructure, particularly in sustainable management of asphalt pavements (as it significant and important part) through planning of preventative maintenance activities while pavement is still in relatively good condition (before it falls into structural problems – problems of insufficient subgrade capacity) and before its replacement is required. Sustainability as important issue is integrated through process of planning maintenance activities within the constrained available resources (financially, temporally, equipment, etc.) while maintaining a satisfactory quality of road infrastructure (technical aspects of maintenance activities on infrastructure elements – road sections, as well its feasibility, etc.) and traffic quality (LOS) for end users (within the area under analysis)."

3.The MCDM methods used in the work are based on the determination of the subjective normalized weights of the criteria and the prioritization of alternatives in the calculations. It seems unusual that the most important formulas of AHP and PROMETHEE methods are not presented in the work.

Thank you very much for this comment. The description of the methods has been added in the paper.

In text: Lines 225-321

4.It is necessary to specify whether the criteria weights were calculated from a single expert or from pairwise matrices completed by a group of experts. In all cases, it is necessary to calculate the eigenvectors “Omega” the largest eigenvalue λmax, and the consistency ratio (C.R.) of each expert completed matrix criteria to show whether the matrix is ​​consistent and whether the transitivity condition has been met. If the criteria weights have been determined from the average of the opinions of the expert group, it is also necessary to prove the consistency of their opinions by calculating, for example, the coefficient of concordance W using the Kendall rank correlation method. These calculations would show that the results of the study are reliable and that the presented methodology could be used by other researchers.

Thank you very much for the comment. The values of eigenvector () and max eigenvector λmax are included in Table 2.

5.The hierarchical structure presented in Figure 3 consists of a goal (the first level), three main criteria (the second level) and three to four sub-criteria (the third level). In Table 2, the weights of the ten sub-criteria for each of the four scenarios are presented as a percentage in a hierarchically unstructured form. It is not clear, by which way were they calculated? (PS. The integer part of the numbers must be separated by a dot, but not a comma).

Thank you very much for the comment. The weights of criteria are calculated using principles of the AHP method and is explained in the paper. Integer part of the numbers is separated by a dot.

6.The calculation data (Table 2, 3, 4 and Fig. 4) are insufficiently analyzed in the text. Fig. 4. is not understandable to the reader because the legends are not visible (they are not legible). In formulas of the final goal function (1) and financial constraint (6) the commas must be replaced by dots. It is not enough sufficient explained how the result was obtained according the functions and the significance of this result both for science and practice was not revealed as well. Is the result worldwide universal, or is it only suitable for the city of Split?

Thank you very much for this comment. Better explanations of the calculation data has been added in the paper. A new Figure 4 has been added. Integer part of the numbers is separated by a dot.
It is not enough sufficient explained how the result was obtained according the functions and the significance of this result both for science and practice was not revealed as well.

In text: “In theoretical terms, there are three types of constraints, two of which are defined as part of this research. Some of them are universally applicable such as this financial constraint, while others refer to spatial analyzes for the purpose of uniform and equitable development of the analyzed area. The first is almost always related to finance as the most important limiting factor. The equations is defined in such a way that on the left side are net flows of road sections (this part refers to alternatives/locations), while the right part is equal to the financial amount defined by the investment plan or decision of the decision-makers. The second type of constraints is related to the spatial-functional segment. These formulas describe a specific constraint arising from the desire to meet strategic goals. Spatial-functional formulas serve to define the constraints that seek to meet the strategic goal of uniformly maintaining the road infrastructure in the entire area of urban infrastructure under analysis. How many formulas will be defined for the spatial-functionals goals depends on the decisions of the decision makers how much detail is needed to allow a uniform approach to road infrastructure maintenance. The third type of formula refers to explicit goals that need to be improved due to plans, regulations, and higher-order goals. When including a formula of this type, locations that are of explicit importance to decision makers are automatically selected. This approach is extremely important in practical terms because it allows the overlap of roads section ranking with the strategies and plans of local governments as well as the preferences of their representatives as ultimate decision- makers. The result is given in the form of annual sets of road sections that need to be repaired.

Within this research, one financial constraint was defined, while for the purpose of uniform development, the research area was divided into four subarea and, in relation to the above, four spatial-functional constraints were defined.”

Is the result worldwide universal, or is it only suitable for the city of Split?

In text: “First of all, the concept is flexible and it is easy and fast to make certain changes depending on the specific task. The changes most often refer to the last level of the goal hierarchical structure (GHS), which refers to the criteria used when comparing variant solutions. It is possible to add new or amend and completely remove existing criteria in order to adapt the concept to the specific task, but it is especially important that decisions are made in the same way as in this process, involving all relevant stakeholders.”

7.Conclusion is formulated due to insufficient disclosure of data obtained by calculations (AHP and PROMETHEE methods). They present the advantages, benefits and possibilities of applying the new DSC concept. However, the specialists of several asphalt pavement maintenance institutions will be able to use it in their activities to a limited extent.

Thank you very much for this comment. The remarks are included in the text in the conclusion.

In text: “The use of multi-criteria methods within the DSC has increased reliability and objectivity in the adoption of annual plans for the maintenance of asphalt pavement constructions. The proposed concept was validated in the area of the city of Split, where a set of 50 alternative solutions (road sections) was defined. The goal was to define a priority ranking of the road sections that need to be repaired. Given that the main limiting factor is the financial nature, and on the other hand it is necessary to meet the condition of uniform development of all parts of the city, restrictions were introduced: one financial limit defining the annual amount in the city budget for maintenance of asphalt pavement and four spatial-functional constraints with respect to the spatial division of the research area. The final result is an annual plan for maintenance of asphalt pavement construction which in this case consists of 11 road sections for research area. The proposed concept is unique and above all easy to use because it allows the simultaneous analysis of a large amount of different data and information with the introduction of stakeholder preferences with different attitudes and desires. In addition to all its advantages, certain disadvantages have been identified that will serve as a basis for future research. The reason for the length of the procedure is primarily in the method of data collection as well as the methods of assessing the asphalt pavement. For the full development of this concept in the direction of its expansion into a system that will also include other elements of road infrastructure, it is first necessary to automate the data collection process, using e.g. road infrastructure safety mapping system using georeferenced video. In order to assess the condition of asphalt pavements, expert systems and neural networks should be included in the future, which would speed up the process on the one hand, while eliminating the subjective character of experts involved in solving a specific task. The proposed concept is flexible, and therefore more widely applicable, whether it is another case study, or another element of road infrastructure. The above can be achieved by structural changes at the last level of GHS (criteria level) that include adding new or amending and completely removing existing criteria in order to adapt the concept to the specific task.”

8.References are not presented in accordance with the requirements of the journal, i. e. the No of vol., issue, pp. and doi. are not marked at all in many publications of Reference list.

Thank you very much for this comment.  References are now corrected.

Reviewer 2 Report

The paper provides an interesting multi-criteria decision-making process for maintenance and asphalt pavement construction. The following observations are minor and tailored to improve the manuscript.

1) The abstract should be written in the past tense and must capture the key findings and implications of the decision-making tool for policy and asphalt construction.

2) Line 60- 90 requires references for sufficient critical writing and justification of the problems.

3) Line 129, the exact number of the three groups of citizens involved in the study should be stated.

4) Line 152 should provide a theoretical background into the working and applications of AHP and PROMETHEE II. I will advise the authors to create a subsection under the methodology for this observation.

5) Line 253-280 containing the formulae requires detailed explanations on their application. The explanations should indicate cases or theoretical examples of how the formulae can be applied for practical use.

6) A section is required for the implications of the findings in practice. An adequate comparison should be made by looking at existing decision support systems for asphalt pavements and associated practical applications. The implications of the findings on government policy must be documented.

7) The limitations and potential for further studies must be stated in the conclusion.

Author Response

Thank you very much for provision of remarks and suggestions. The authors hope that corrections and further explanations which are provided according to your comments and suggestions improved quality of the manuscript up to the satisfactory level it can be accepted for publication in Sustainability.

1.The abstract should be written in the past tense and must capture the key findings and implications of the decision-making tool for policy and asphalt construction.

Thank you very much for the comment. The abstract is written in the past tense and the key findings and implications of the proposed decision support concept are highlighted.

2.Line 60- 90 requires references for sufficient critical writing and justification of the problems.

Thank you very much for the comment. Authors included references for critical writing in lines 57-111.

3.Line 129, the exact number of the three groups of citizens involved in the study should be stated.

Thank you very much for this comment. The exact number of the stakeholders in every group has been added in the text.

In text: „Application of the concept begins with the determination of research problem including determination of the study area and selection the relevant stakeholders who are divided into three groups:
- Citizen representatives (9 representatives of city districts),
- Experts in transport engineering (2 engineers from the utility company in charge of road maintenance, 2 engineers from the company that performs road maintenance works, 3 construction experts from the Faculty of Civil Engineering, Architecture and Geodesy, 2 engineers from the Faculty of Civil Engineering, Architecture and Geodesy, 3 experts in methodology), and
- Government representatives (representative of the utility company in charge of maintenance, deputy mayor in charge of infrastructure and head of the department in charge of construction and infrastructure). Each group came up with common weighting criteria. The experts in charge of the methodology were tasked with explaining to all other stakeholders how the criteria were compared. Finally, the weight of the criteria were determined by the AHP method [8].”

4.Line 152 should provide a theoretical background into the working and applications of AHP and PROMETHEE II. I will advise the authors to create a subsection under the methodology for this observation.

Thank you very much for this comment. The description of the methods has been added in the paper.

In text: Lines 225-321

5.Line 253-280 containing the formulae requires detailed explanations on their application. The explanations should indicate cases or theoretical examples of how the formulae can be applied for practical use.

Thank you very much for this comment. The detailed explanation has been added in the text.

In text: “In theoretical terms, there are three types of constraints, two of which are defined as part of this research. Some of them are universally applicable such as this financial constraint, while others refer to spatial analyzes for the purpose of uniform and equitable development of the analyzed area. The first is almost always related to finance as the most important limiting factor. The equations is defined in such a way that on the left side are net flows of road sections (this part refers to alternatives/locations), while the right part is equal to the financial amount defined by the investment plan or decision of the decision-makers. The second type of constraints is related to the spatial-functional segment. These formulas describe a specific constraint arising from the desire to meet strategic goals. Spatial-functional formulas serve to define the constraints that seek to meet the strategic goal of uniformly maintaining the road infrastructure in the entire area of urban infrastructure under analysis. How many formulas will be defined for the spatial-functionals goals depends on the decisions of the decision makers how much detail is needed to allow a uniform approach to road infrastructure maintenance. The third type of formula refers to explicit goals that need to be improved due to plans, regulations, and higher-order goals. When including a formula of this type, locations that are of explicit importance to decision makers are automatically selected. This approach is extremely important in practical terms because it allows the overlap of roads section ranking with the strategies and plans of local governments as well as the preferences of their representatives as ultimate decision- makers. The result is given in the form of annual sets of road sections that need to be repaired.

Within this research, one financial constraint was defined, while for the purpose of uniform development, the research area was divided into four subarea and, in relation to the above, four spatial-functional constraints were defined.”

6.A section is required for the implications of the findings in practice. An adequate comparison should be made by looking at existing decision support systems for asphalt pavements and associated practical applications. The implications of the findings on government policy must be documented.

Thank you very much for the comment. The comparison of the proposed study with existing decision making models for asphalt pavements and practical applications is given in the Conclusion.

In text: “Comparing with the existing decision making models such as [11] and [12], it is obvious that these methods did not used the multicriterial approach to evaluate and analyze each alternative according to the defined criteria. No selection and division of relevant stakeholders was made, as a crucial factor in the whole decision making process. Finally, the actions are mentioned but their priority in the maintenance is not demonstrated and included in the overall outranking process. As for the studies [13] and [14], only two activities are included in the asphalt pavement maintenance under the Rough Set Theory in the decision making procedure. The first evaluated the effect of failure on the intended function of the pavement, and the second is a visual inspection determining the condition of pavement and the problems that are causing this condition. The methodologies are only based on the condition of the pavement, and no additional attention was given to other parameters that also have a notable impact on the whole construction, such as capacity, stability, safety, functionality and costs.”

7.The limitations and potential for further studies must be stated in the conclusion.

Thank you very much for this comment. The remarks are included in the text in the conclusion.

In text: “The use of multi-criteria methods within the DSC has increased reliability and objectivity in the adoption of annual plans for the maintenance of asphalt pavement constructions. The proposed concept was validated in the area of the city of Split, where a set of 50 alternative solutions (road sections) was defined. The goal was to define a priority ranking of the road sections that need to be repaired. Given that the main limiting factor is the financial nature, and on the other hand it is necessary to meet the condition of uniform development of all parts of the city, restrictions were introduced: one financial limit defining the annual amount in the city budget for maintenance of asphalt pavement and four spatial-functional constraints with respect to the spatial division of the research area. The final result is an annual plan for maintenance of asphalt pavement construction which in this case consists of 11 road sections for research area. The proposed concept is unique and above all easy to use because it allows the simultaneous analysis of a large amount of different data and information with the introduction of stakeholder preferences with different attitudes and desires. In addition to all its advantages, certain disadvantages have been identified that will serve as a basis for future research. The reason for the length of the procedure is primarily in the method of data collection as well as the methods of assessing the asphalt pavement. For the full development of this concept in the direction of its expansion into a system that will also include other elements of road infrastructure, it is first necessary to automate the data collection process, using e.g. road infrastructure safety mapping system using georeferenced video. In order to assess the condition of asphalt pavements, expert systems and neural networks should be included in the future, which would speed up the process on the one hand, while eliminating the subjective character of experts involved in solving a specific task. The proposed concept is flexible, and therefore more widely applicable, whether it is another case study, or another element of road infrastructure. The above can be achieved by structural changes at the last level of GHS (criteria level) that include adding new or amending and completely removing existing criteria in order to adapt the concept to the specific task.”

Round 2

Reviewer 1 Report

The authors of the article supplemented and provided more detailed comments on the methods used in the work. In addition, the authors performed a deeper discussion of the research results obtained and their application. This contributed to a significant improvement in the quality of the peer-reviewed article. The basic formulas of the PROMETHEE and AHP methods were supplemented.

However, some shortcomings of the article should be noted:

  1. Should be “…AHP (Analytic Hierarchy Process) method. (line 16, as on lines 59 and 145).
  2. Not all literature sources have indications: vol., issues and pages.
  3. Table 2 should be presented according to the layout of template, in which the C8…C10 would be visible.
  4. Line 248 “…where m is the number of considered evaluation criteria” (Eq.2). Line 266 “… n is the matrix dimension” (Eq. 3). Do m and n show different values?
  5. I recommend to provide (display) at least by one expert filled matrix y (pairwise matrix) of criteria for pairwise comparison and the calculated values of the criteria weight (eigenvectors ωi).
  6. Recommended References for supplement: Transportation Research Record 2093(2009):12-24; Journal of Cleaner Production (2009) 17(:2): 283-296; Transport (2011), 26(1):20-34; Journal of Testing and Evaluation (2016), 44(1): 89-101.

Author Response

1.Should be “…AHP (Analytic Hierarchy Process) method. (line 16, as on lines 59 and 145).

Thank you very much for the comment. The suggested is entered in the manuscript. 

2.Not all literature sources have indications: vol., issues and pages.

145).

Thank you very much for the comment. The suggested is entered in the manuscript. 

3.Table 2 should be presented according to the layout of the template, in which the C8…C10 would be visible.

Thank you very much for the comment. The suggested is entered in the manuscript. 

4.Line 248 “…where m is the number of considered evaluation criteria” (Eq.2). Line 266 “… n is the matrix dimension” (Eq. 3). Do m and n show different values?

Thank you very much for the comment. The suggested is entered in the manuscript. 

5.I recommend to provide (display) at least by one expert filled matrix y (pairwise matrix) of criteria for pairwise comparison and the calculated values of the criteria weight (eigenvectors ωi).

Thank you very much for the comment. Authors insert one more Table where criteria weights given by one stakeholder form citizen representatives (SC1) are presented.

6.Recommended References for supplement: Transportation Research Record 2093(2009):12-24; Journal of Cleaner Production (2009) 17(:2): 283-296; Transport (2011), 26(1):20-34; Journal of Testing and Evaluation (2016), 44(1): 89-101.

Thank you very much for the comment. Authors entered recommended references in the manuscript, as suggested.
